# Urinary minerals excretion among primary schoolchildren in Dubai—United Arab Emirates

Rola Al Ghali[1], Carla El-Mallah[2], Omar Obeid[2], Ola El-Saleh[3], Linda Smail[4], Dalia Haroun[1] *

1 Department of Public Health and Nutrition, College of Natural and Health Sciences, Zayed University, Dubai, United Arab Emirates, 2 Department of Nutrition and Food Science. Faculty of Agricultural and Food Sciences, American University of Beirut, Beirut, Lebanon, 3 Department of Medicine, School of Public Health, Imperial College London, London, United Kingdom, 4 Department of Mathematics and Statistics, College of Natural and Health Sciences, Zayed University, Dubai, United Arab Emirates

* Dalia.haroun@zu.ac.ae

## Abstract

### Introduction

Urinary excretion of calcium (Ca), magnesium (Mg), phosphorus (P), iodine and fluoride is used to assess their statuses and/or the existence of metabolic abnormalities. In the United Arab Emirates (UAE), the urinary concentration of these minerals among children have not been documented.

### Materials and methods

A cross-sectional study, including 593 subjects (232 boys and 361 girls), was conducted among healthy 6 to 11-year-old Emirati children living in Dubai. Non-fasting morning urine samples and anthropometrical measurements were collected and analyzed. Results were expressed as per mg of creatinine (Cr).

### Results

On average, estimated Cr excretion was 17.88±3.12 mg/kg/d. Mean urinary Ca/Cr, Mg/Cr and P/Cr excretions were 0.08±0.07 mg/mg, 0.09±0.04 mg/mg, and 0.57±0.26 mg/mg respectively. Urinary excretion of Ca, Mg and P were found to decrease as age increased. Urinary excretion and predicted intake of fluoride were lower than 0.05 mg/kg body weight per day. Surprisingly, more than 50% of the children were found to have urinary iodine excretion level above adequate.

### Conclusion

The Emirati schoolchildren had comparable levels of urinary Ca, Mg and P excretion to other countries. The 95% percentile allows the use of the current data as a reference value for the detection of mineral abnormalities. Fluoride excretion implies that Emirati children are at low risk of fluorosis. The level of urinary iodine excretion is slightly higher than

**Data Availability Statement:** All relevant data are within the manuscript.

**Funding:** DH, OO and RA received funding from Zayed University to conduct the study (R15116

(The funders had no role in study design, data collection and analysis, decision to publish, or preparation of the manuscript.

**Competing interests:** The authors have declared that no competing interests exist.

**Abbreviations:** Ca, Calcium; Mg, Magnesium; P, Phosphorus; UAE, United Arab Emirates; Cr, Creatinine; UIC, Urinary Iodine Concentration; MoE, Ministry of Education; ZU, Zayed University; MoHAP, Ministry of Health and Prevention; BMI, Body Mass Index; CDC, Centers for Disease Control and Prevention; EDTA, Ethylene dinitrilotetracetic acid; HCl, Hydrochloric acid; DUFE, Daily urinary fluoride excretion; TISAB, Total ionic strength activity buffer solution; TDFI, Total daily fluoride intake; SD, Standard deviation; LSD, Least Significant Difference; WHO, World Health Organization; AI, Adequate intake; UL, Upper limit; IOM, Institute of Medicine; GCC, GSO: Gulf Corporation Council Standardization Organization; AUB, American University of Beirut.

recommended and requires close monitoring of the process of salt iodization to avoid the harmful impact of iodine overconsumption.

## Introduction

Childhood is the most rapid and fundamental period of development. To ensure optimal and profound growth, a diet with adequate nutrients and minerals, naming Ca, Mg, P, iodine and fluoride, is essential during this stage. Unbalanced status of these minerals during childhood can have long-term implications. Ca, Mg, and P are macrominerals that are known, among other functions, to be related to bones and muscle contraction [1, 2]. Iodine is linked to proper thyroid and cognitive functions [3–5], while fluoride is a key element in dental and oral health [6]. Therefore, imbalances in the status of these minerals, whether due to dietary intake, body retention or excretion, can be multifactorial, but their presence indicates possible metabolic anomalies. The kidneys, with other organs and hormones, play an important role in the homeostasis of Ca, Mg, P, fluoride, and iodine [1, 5, 7, 8]. Hence, urinary excretion is a good reflector and indicator of the status of these minerals. This can be evaluated by either collecting a 24-hour urine sample or using a spot urine test. Although 24-hour urine collection is the gold standard, it is cumbersome, expensive and labor-intensive. Therefore, a non-fasting spot urine test is a more feasible, reliable screening tool and is commonly used to determine mineral excretions in population-based studies [9], especially among young populations [10–16]. Most mineral excretions are frequently expressed in a ratio to Cr excretion (solute/Cr). Cr ratios are used to adjust for all inconstant variables like dehydration or over hydration when mineral excretions are studied [17].

To evaluate mineral metabolic disorders among children, mineral excretion levels should be compared to normal reference values. While iodine excretion has well-defined cutoffs among the pediatric population [18], many other minerals still have not. Factors like genetics, race, age, climate and dietary intake [10–14, 19, 20], in addition to the mineral content of drinking water [21] can affect the urinary mineral excretion.

To the best of our knowledge, the excretion levels of Ca, Mg, P, iodine, and fluoride have not been studied, and reference values have not been reported among school-aged children in Dubai—UAE. The aim of this study is to assess iodine and fluoride status and to determine the reference values for urinary Mg/Cr, Ca/Cr, and P/Cr of healthy 6 to 11-year-old Emirati children. This will help in exploring possible attributes to the young Emirati population's health and establishing national references for potential future use to diagnose any health-related anomalies in mineral metabolism.

## Materials and methods

### Study design

This is a cross-sectional study investigating the urinary excretion of Ca, Mg, P, iodine and fluoride among school-aged children attending primary public schools in Dubai, UAE. Data were collected between May and July 2017.

### Study population and sampling

A comprehensive list including all Dubai primary public schools (28 schools), the number of enrolled students (11,147) and their demographics was provided by the Ministry of Education

(MoE) targeting 5% of the total population. Public schools in Dubai are sex-segregated and consist mainly of Emirati students. Out of the list of 28 schools, 23 agreed to take part in the study, and a total of 612 students (241 boys and 371 girls) aged between 6 and 11 years (grades 1–5) were recruited (51% response rate). Students were stratified based on sex and age. Participants were selected randomly from a list including all enrolled students via Research Randomizer (Version 4.0), while ensuring the number of students selected from each school is proportional to the enrolment size and demographic characteristics of the population. All ethical approvals and clearances needed were granted from Zayed University (ZU) ethical committee, the MoE and the Ministry of Health and Prevention (MoHAP) (MOHAP/DXB/SUBC/NO-17, 2016). Prior to data collection and school visits, consent forms were sent to the guardians of the selected students along with a data collection sheet to identify the child's demographics and medical history. Written informed consent was obtained from parents and verbal assent was sought from the participating students. From a pool of 612 students, 593 were included. The 19 excluded participants included 4 non-Emiratis, 2 drop-out students, 2 students with a known diagnosis of hypertension and diabetes, 3 students with renal disease, and 8 students with missing data such as age, weight, height and nationality.

### Data collection

All students' assessments were conducted by research assistants/licensed school nurses who have been adequately trained by the MoHAP on the standardized protocols for quality control and assurance.

### Anthropometry

Standing height was measured to the nearest 0.1 cm using a portable stadiometer (Charder, HM-200P). Whilst weight was measured in light clothing to the nearest 0.1 kg using a calibrated digital scale (TANITA, BC-420MA). Body Mass Index (BMI) was calculated as the ratio of weight (kg)/height (m)$^2$. BMI values were converted to BMI z-scores based on age and sex as per the 2000 Centers for Disease Control and Prevention (CDC) growth charts [22].

### Urine samples

Spot urine samples were collected from students during school hours between 09:00 and 11:00 am. Screw-capped urine sterile tubes (Sarstedt, 62.476.028, Numbrecht, Germany) labelled with the student's identification number were utilized. Samples were then aliquoted into 2 ml microtubes on the same day of collection and stored at -20˚C [23] in the freezer at ZU Science Laboratory. Urine samples for fluoride analysis were preserved in ethylene dinitrilotetracetic acid (EDTA). All samples were shipped to the American University of Beirut (AUB), Lebanon.

### Analysis of minerals

Testing for Ca, Mg, and P, urine samples were defrosted and acidified with hydrochloric acid (HCl) to prevent mineral precipitation [19]. Sixty µl of 3N HCl were added to 5 ml of urine to bring pH of the specimens between 3 and 4. Ca, Mg, and P were determined by the absorbance of the colored complex with formazan, the colorimetric reactions with Arsenazo III dye, and the ammonium molybdate, respectively, all using VITROS 350 analyzer (Ortho Clinical Diagnostics, Johnson and Johnson, Buckinghamshire, United Kingdom). Urinary iodine was measured using a modification of the Sandell–Kolthoff method at CDC-certified Ensuring the Quality of Urinary Iodine Procedures (EQUIP) laboratory of AUB. Before fluoride reading, 10 ml of total ionic strength activity buffer solution (TISAB) were added, and urinary fluoride

was measured using fluoride ion-selective electrode (Thermo Scientific, Beverly, MA 01915, USA). Urinary Cr was measured using two-point rate procedure using Vitros 350 analyzer. Cr excretion in mg/kg/day was calculated using the equation of Remer et al. [23]. The 24-h fluoride excretion was estimated based on 24-h Cr excretion. Referring to the regression data of Villa et al., 2010 that correlate daily urinary fluoride excretion (DUFE) with the total daily fluoride intake (TDFI), estimated daily fluoride intake was calculated following this equation: DUFE = 0.03 + (0.35 x TDFI) [24].

## Internal control

Internal controls were performed after each run to monitor the accuracy of the readings. The coefficients of variance of the "between-runs" measurements of Ca, Mg, P, iodine, fluoride, and Cr were 1.94%, 1.29%, 1.19%, 8.3%, 2%, and 1.42%, respectively.

## Statistical analysis

Statistical analysis was performed using SPSS 26 (IBM SPSS). Statistical significance was set at a $P$-value $<0.05$. Since the sample size is large enough, parametric tests were performed, assuming normality based on the central limit theorem.

Results were presented as means ± standard deviation (SD) for quantitative variables and percent for categorical variables. Data were stratified by sex and/or by six age groups (6–6.9, 7–7.9, 8–8.9, 9–9.9, 10–10.9 and 11–11.9 years). A two-sample independent $t$-test was used to identify differences by sex and one-way ANOVA with Least Significant Difference (LSD) post-hoc tests to evaluate differences among age groups. The numbers reported for each mineral differed due to insufficient urine quantities to run analysis or duplicates for Ca (61), Mg (61), P (63), iodine (63) and fluoride (90). Hence, the analyzed sample size (n) for Ca and Mg was 532; 530 for P and iodine; and 503 for fluoride.

## Results

This study recruited 593 primary school Emirati children, 232 boys (39%) and 361 girls (61%), aged between 6 and 11.9 years. The majority of the population had normal BMI (59.4%), with a quarter being overweight or obese. There were no significant differences across sex for both age and height, while girls had higher weight and BMI readings than boys ($P <0.05$). On average, estimated Cr excretion was 17.88±3.12 mg/kg/d. Boys had a marginal higher Cr excretion than girls ($P = 0.045$) with estimated levels of 18.19±2.97 mg/kg/d and 17.68±3.20 mg/kg/d among boys and girls, respectively (Table 1).

Total mean Ca/Cr was 0.08±0.07 mg/mg, Mg/Cr was 0.09±0.04 mg/mg, and P/Cr was 0.57 ±0.26 mg/mg (Table 2). In boys, the mean Ca/Cr ($P <0.001$), Mg/Cr (p = 0.015) and Ca/P (P = 0.004), ratios were singnifically higher than that of girls. While, P/Cr was similar between the sexes. On the other hand, mean fluoride to Cr (F/Cr) ratio was 0.67±0.58 mg/g with an estimated 24 hour fluoride (24-h F) intake 0.03±0.03 mg/kg/d and these were similar between boys and girls. Similarly, urinary iodine concentration (UIC) 224.24±108.76 µg/l was similar between the two sexes (Table 3).

Further analysis was done to detect the effect of age on the excretion of Cr and mineral to Cr ratios. Age showed no substantial influence on the excretion of both Cr and Ca/P ($P >0.05$), while statistical significance was reported in mean Ca/Cr ($P = 0.004$), Mg/Cr ($P <0.001$), and P/Cr ($P <0.001$); all showing a decreasing trend between the ages of 6 and 12 years. Interestingly, post-hoc tests showed that Ca/Cr ratio (mg/mg) among 7–7.9 years was significantly different than all other age groups ($P <0.05$). Mg/Cr ratio (mg/mg) showed a similar trend among the 7–7.9-year-olds with ($P <0.05$) except with 6–6.9 years ($P >0.05$), in

**Table 1. Anthropometric characteristics of the sample population.**

|  | Total (N = 593) | Boys (N = 232) | Girls (N = 361) | P-value |
|---|---|---|---|---|
| Age (years) | 8.7±1.4 | 8.6±1.4 | 8.7±1.3 | 0.403 |
| Weight (kg) | 30.1±10.2 | 28.9±8.9 | 30.9±10.8 | 0.013 |
| Height (cm) | 130.7±9.6 | 130.2±8.4 | 131.1±10.2 | 0.219 |
| BMI z-score | 0.08±1.47 | -0.24±1.47 | 0.02±1.45 | 0.031 |
| Cr (mg/kg/d) | 17.88±3.12 | 18.19±3.00 | 17.68±3.20 | 0.045 |

All values are reported as mean ± SD. Significance is set at *P*-value < 0.05.

addition to significant differences between 10–10.9 years and both 6–6.9 years and 8–8.9 years (*P* <0.05). P/Cr ratio (mg/mg) showed significant differences between age group brackets of 6–6.9 and 9–9.9 years (*P* <0.05); 7–7.9 years and all other age groups (*P* <0.05) except for 6–6.9 years (Table 4).

As for fluoride and iodine, the parameters that showed significant differences with age were F/Cr ratio (mg/g), estimated 24-h F intake (mg/kg/d) and I/Cr (μg/g) with *P*-values less than 0.05. Post-hoc tests showed significant differences (*P* <0.05) in fluoride excretion and estimated intake when comparing the 6–6.9 year-olds with 7–7.9, 8–8.9 and 10–10.9 year-old children, in addition to a significant difference between 9–9.9 and 10–10.9 years. I/Cr ratio had a *P* <0.05 between ages 6–6.9 and both 10–10.9 and 11–11.9 years; 7–7.9 compared to both 10–10.9 and 11–11.9 years; and age groups 9–9.9 and 10–10.9 years. The highest reported means for F/Cr ratio (mg/mg), estimated 24-h fluoride intake (mg/kg/d) and I/Cr ratio (mg/mg) were among the 6–6.9 year category. These means decreased as age increased (Table 5). Supplementary analysis was done to detect the effect of sex across these age groups. Despite the fact that F/Cr and I/Cr were statistically different between age groups, no sex differences were reported.

In reference to the Institute of Medicine (IOM), fluoride adequate intake (AI) is set at 0.05–0.07 mg/kg body weight and 0.1 mg/kg as the tolerable upper intake limit (UL) [25]. Our findings show that only 14.5% of the study population met the AI, with more than 80% consuming less. UL was exceeded by 4% of our sample.

**Table 2. Urinary Ca/Cr, Mg/Cr, P/Cr, and Ca/P ratios of elementary Emirati schoolchildren.**

| Parameter<br>Unit | N | Total | N | Boys | N | Girls | P-value |
|---|---|---|---|---|---|---|---|
| Ca/Cr | 532 |  | 212 |  | 320 |  | <0.001 |
| mg/mg |  | 0.08±0.07 |  | 0.09±0.08 |  | 0.07±0.06 |  |
| Mg/Cr | 532 |  | 210 |  | 322 |  | 0.015 |
| mg/mg |  | 0.09±0.04 |  | 0.10±0.04 |  | 0.09±0.04 |  |
| P/Cr | 530 |  | 210 |  | 320 |  | 0.207 |
| mg/mg |  | 0.57±0.26 |  | 0.58±0.27 |  | 0.55±0.25 |  |
| Ca/P | 530 |  | 210 |  | 320 |  | 0.004 |
| mg/mg |  | 0.15±0.15 |  | 0.17±0.16 |  | 0.14±0.14 |  |

For Ca 1 mg/dl is equivalent to 0.250 mmol/l; Mg 1 mg/dl is equivalent to 0.412 mmol/l; P 1 mg/dl is equivalent to 0.321 mmol/l; and Cr 1 mg/dl is equivalent to 0.088 mmol/l.

All values are represented as mean ±SD.

*t*-test is used to compare means among sexes and significance is set at P-value <0.05.

**Table 3. Fluoride and Iodine statuses of elementary Emirati schoolchildren.**

| Parameter | N | Total | N | Boys | N | Girls | P-value |
|---|---|---|---|---|---|---|---|
| Unit | | | | | | | |
| F/Cr | 503 | | 202 | | 301 | | 0.779 |
| mg/g | | 0.67±0.58 | | 0.68±0.60 | | 0.67±0.58 | |
| Estimated 24-h F intake | 503 | | 202 | | 301 | | 0.627 |
| mg/kg/d | | 0.03±0.03 | | 0.03±0.03 | | 0.03±0.03 | |
| UIC | 530 | | 211 | | 319 | | 0.481 |
| µg/l | | 224.24±108.76 | | 220.22±102.52 | | 226.93±112.81 | |
| I/Cr | 530 | | 211 | | 319 | | 0.891 |
| µg/g | | 219.50±126.92 | | 217.95±125.77 | | 220.53±127.85 | |

For F 1 mg/dl is equivalent to 0.526 mmol/l and I1 µg/l is equivalent to 7.88 nmol/l.

All values are represented as mean ±SD.

*t*-test is used to compare means among sexes and significance is set at P-value <0.05.

Although individual urinary iodine excretion can vary on a daily basis and UIC is useful as a good simple measure for populations [18], it is worth noting that more than 50% of the Emirati population were above the adequate level and about 10% were found deficient.

## Discussion

This study aims at assessing iodine and fluoride status, as well as the urinary excretion levels of Mg, Ca, and P per mg of Cr among healthy 6 to 11-year-old Emirati children living in Dubai. These levels could be used to determine the potential existence of mineral-related metabolic abnormalities when excretion exceeds the 95[th] percentile. Data were expressed as per mg of Cr, which is known to be produced at an almost constant rate in the body [26] and its excretion is constant within individuals [27]. Cr highly reflects body metabolism in healthy conditions and is related to lean body mass [23]. Studies on Cr excretion among children are scarce and no exact reference values exist [23], but it is assumed that the average Cr per body weight among children before puberty is around 18 mg/kg/d [28]. Our results showed a similar value in which mean Cr excretion among Emirati children was 17.88±3.12 mg/kg/day, and this further validates the urinary assessment of the varied minerals. Urinary creatine was comparable

**Table 4. Urinary Ca/Cr, Mg/Cr, P/Cr, and Ca/P ratios among elementary Emirati schoolchildren of varied age groups.**

| Age | N | Cr | N | Ca/Cr | N | Mg/Cr | N | P/Cr | N | Ca/P |
|---|---|---|---|---|---|---|---|---|---|---|
| Years | 593 | mg/kg/d | 532 | mg/mg | 532 | mg/mg | 530 | mg/mg | 530 | mg/mg |
| 6–6.9 | 62 | 17.65±2.88 | 52 | 0.08±0.08 | 53 | 0.10±0.04 | 54 | 0.59±0.28 | 54 | 0.13±0.12 |
| 7–7.9 | 117 | 17.76±2.61 | 97 | 0.10±0.08 | 98 | 0.11±0.04 | 96 | 0.67±0.27 | 96 | 0.15±0.15 |
| 8–8.9 | 143 | 18.01±3.10 | 126 | 0.08±0.07 | 125 | 0.09±0.04 | 126 | 0.57±0.25 | 126 | 0.14±0.13 |
| 9–9.9 | 126 | 17.80±3.36 | 119 | 0.08±0.07 | 118 | 0.09±0.04 | 117 | 0.51±0.23 | 117 | 0.16±0.15 |
| 10–10.9 | 117 | 18.02±3.41 | 112 | 0.07±0.06 | 111 | 0.08±0.03 | 110 | 0.53±0.26 | 110 | 0.15±0.16 |
| 11–11.9 | 28 | 18.05±3.47 | 26 | 0.06±0.05 | 27 | 0.08±0.04 | 27 | 0.52±0.24 | 27 | 0.16±0.18 |
| P-value | | 0.944 | | 0.014 | | <0.001 | | <0.001 | | 0.955 |

For Ca 1 mg/dl is equivalent to 0.250 mmol/l; Mg 1 mg/dl is equivalent to 0.412 mmol/l; P 1 mg/dl is equivalent to 0.321 mmol/l; and Cr 1 mg/dl is equivalent to 0.088 mmol/l.

All values are represented as mean ±SD.

One-way ANOVA with LSD post-hoc tests is used to compare means among age groups and significance is set at P-value <0.05.

**Table 5. Fluoride and Iodine statuses among elementary Emirati schoolchildren of varied age groups.**

| Age | N | F/Cr | Estimated 24-h F intake | N | UIC | N | I/Cr |
|---|---|---|---|---|---|---|---|
| Years | 503 | mg/g | mg/kg/d | 530 | µg/l | 530 | µg/g |
| 6–6.9 | 51 | 0.90±0.69 | 0.04±0.04 | 53 | 216.72±112.84 | 53 | 245.75±155.71 |
| 7–7.9 | 88 | 0.67±0.56 | 0.03±0.03 | 96 | 216.22±110.55 | 96 | 239.22±151.01 |
| 8–8.9 | 120 | 0.67±0.52 | 0.03±0.03 | 126 | 224.63±113.67 | 126 | 221.06±117.86 |
| 9–9.9 | 113 | 0.77±0.71 | 0.03±0.04 | 120 | 232.17±110.77 | 120 | 225.41±130.82 |
| 10–10.9 | 104 | 0.52±0.43 | 0.03±0.02 | 109 | 23030±104.13 | 109 | 190.25±94.64 |
| 11–11.9 | 27 | 0.65±0.46 | 0.03±0.03 | 26 | 206.76±77.60 | 26 | 181.03±75.73 |
| P-value | | 0.006 | 0.026 | | 0.783 | | 0.022 |

For F 1 mg/dl is equivalent to 0.526 mmol/l and I1 µg/l is equivalent to 7.88 nmol/l.

All values are represented as mean ±SD.

One-way ANOVA with LSD post-hoc tests is used to compare means among age groups and significance is set at P-value <0.05.

to that of other populations [29] and was found to be higher among boys as compared to girls [30].

Our data show that urinary Ca/Cr ratio of Emirati children was close to that of American Caucasians [31], and Lebanese [29]. However, the mean Ca/Cr ratio was slightly (16% to 35%) lower than the Turkish [32], German [23], Korean [33], Argentinian [12] and Thai children [16], while urinary Ca/Cr excretion of Emirati children was higher than that of Iranian [10], African Americans [31], Swedish [14] and Taiwanese [11] children. The observed lower urinary Ca/Cr excretion among girls was not in line with others [9–12, 29, 32], and may relate to a difference in menarcheal age between the varied populations. Menarcheal age is associated with high Ca accretion and is around the age of 12.5 years among Emirati girls [34]. In support, urinary Ca/Cr excretion was found to decrease as the age get closer to 12 years. This decrease is expected after the age of 10 [10, 11] due to the increased bone mineral deposition, especially older than 12 years [35, 36]. Similar results for Ca/Cr were reported in Switzerland [13], Sweden [14] and Iran [15], where Ca/Cr ratio decreased with age. Due to inter-individual variabilities, the literature has reported wide distribution of mineral excretion levels, reflected by the SD. The SD for Ca/Cr ratio varied between 42.8 and 112.8% [10–12, 14, 28, 29, 33]. In our study, the SD for Ca/Cr ratio was 89.6%, a good percentage was on a positive end with high Ca excretion and a significant number had lower Ca/Cr excretions. In fact, the dietary patterns are far from being uniform and can change with multiple factors such as demographic, economic, and parents' education [37, 38]. Dietary pattern of Emirati adolescents and children was reported to shift towards a higher consumption of sugar sweetened beverages (sodas and fruit drinks) and less consumption of milk [39, 40]; and this is paralleled by low vitamin D status [41–43]. The aforementioned factors may partially explain the relatively low Ca excretion rates among some children. On the other hand, the potential existence of metabolic abnormalities (e.g. hypercalciuria) can be detected by determining the UL or the 95[th] percentile of minerals excretion. In our study, the 95[th] percentile of Ca/Cr was 0.22 mg/mg; this value was slightly lower than that of other populations [12, 16, 29, 32, 44, 45]. Our findings argue against the suggestion that children living in desert areas, like the UAE, have higher urinary Ca excretion compared to those who live near the coast [46]. In relation to this, references from the literature stated that an indication of hypercalciuria is a fasting [12, 15] or casual [32, 47] Ca/Cr ratio that is higher than 0.21 mg/mg and a random non-fasting reading of 0.27 mg/mg [48]. Consequently, an estimated incidence of hypercalciuria among Emirati children would be 6.58% and 3.38% using a cut-off of 0.21 mg/mg [32, 47] and 0.248 mg/mg [45],

respectively. The percentage of hypercalciuria among Emirati children was lower than those reported among children of comparable age from different countries [29, 47].

Mean urinary Mg/Cr excretion of our study population was 0.09±0.04 mg/mg, a value higher than that of Iranian [10] and Taiwanese [11] children, and lower than that of the Lebanese [29], British [28] and Americans [49]. The relation between urinary Mg/Cr excretion and sex is far from clear. It was reported to be higher among boys by our finding, lower among boys by some [10, 11], while others reported no relationship [13, 21, 29]. Additionally, the observed inverse relation between urinary Mg/Cr excretion and age was reported by some [10, 11], but not all [29]. Whatever, the pattern of changes in Mg excretion with age seems to mimic that of Ca.

On the other hand, the SD reported for Mg/Cr ratio in our results was 50%, a value comparable to that of other studies [10, 11, 28, 29]. The 95[th] percentile of Mg/Cr ratio was 0.17 mg/mg, a value lower than that of Lebanese [29], Spanish [44] and Swiss [13] children.

Urinary P/Cr excretion was close to that of the Lebanese [29] and Iranian [10] children, but lower than that of children living in Florida [50]. In line with others, P/Cr ratio excretion was independent of sex [10, 21, 29], though a higher P/Cr excretion among boys was reported by some [7]. Moreover, the observed decline in P/Cr excretion with age was similar to other findings from Lebanon [29], Iran [10], Switzerland [13] and Spain [44]. Urinary Ca/P ratio among the Emirati children was significantly higher among boys and did not change with age. The observed SD for P/Cr of 46% was within the reported range (11.1% and 60%) [10, 11, 28, 29]. The estimated 95[th] percentile of P/Cr ratio (1.03 mg/mg) was lower than the Lebanese [29] and Spanish [44] children, but similar to what came in Kompani et al. results in Iran (1.01 mg/mg) [21].

Furthermore, about 90% of dietary iodine is known to be excreted in the urine, expressed as UIC [18], and this makes it a reliable marker of iodine intake [51, 52]. Our study population had a mean UIC of 224.24±108.76 μg/l and a median of 208.59 μg/l (151.66–288.98 μg/l) or 20.86 μg/dl. Using UIC of school-aged children (6 years or older) as indicator, the study population was slightly above the adequate level which is 100–199 μg/l, posing a slight risk of more than adequate iodine intake [18]. Compared to other populations, the Emirati children's UIC was higher than that of Switzerland (median UIC was 14 μg/dl) [53], Portugal (12.9 μg/dl) [54], Israel (83 μg/l) [55], and Lebanon (66.00 μg/l) [56].

In 1994, thyroid gland disorders among school-aged children in the UAE exceeded 40%, thereafter in 1999, around 67% Emirati children were found not to have sufficient dietary iodine [57]. In response to the initiative of eliminating iodine deficiency, the UAE has started the implementation of the salt iodization program in 2007 in harmony with the Gulf Corporation Council Standardization Organization (GCC GSO) requirements [58, 59], that were updated in 2010 and require the addition of 15–40 ppm of iodine in salt (as sodium and potassium iodides or iodates) [60]. The level of fortification was very close to the internationally acknowledge range, recommended by the WHO (20 to 40 ppm) [18]. In 2009, a significant drop in the enlargement of the thyroid gland from 40% in 1994 to 8.2% in 2009 was reported [57]. In addition, this was accompanied by an increase in the rate of iodized salt utilization among the population from 6.5% in 1994 to 94.1% in 2009 [57]. At the same time, 41.7% of people were reported to have adequate iodine excretion, though only 7.5% of the used salt had iodine concentrations within the recommended international standards (15–40 g iodine/kgs salt) [57]. In our study, we analyzed 10 of the commonly available and used salt brands in the UAE for iodine. Iodine content in mg/kg (ppm) varied from 10.3 ppm to 208 ppm, with 60% of the samples had higher iodine than GSS GSO recommendations [60]. This high level of iodine content is likely to have contributed to the detected slightly high urinary excretion. Therefore, a close monitoring of salt iodization is needed to prevent the harmful impact of

high iodine intake. Hence, policy makers might need to strengthen the monitoring process at the industrial level to ensure abidance by the recommended levels.

Urinary fluoride measurement is the most useful biomarker to assess fluoride status [24, 61] as fluoride excretion is an indicator of fluoride intake [62]. AI of fluoride falls within the range of 0.05–0.07 mg/kg body/day weight in children less than 12 years of age, while UL is set at 0.1 mg/kg body weight/day [63, 64]. About 81.5% of children had estimated intake below 0.05 mg/kg boy weight per day and no one had an intake above the UL. The estimated intake of 0.032±0.031 mg/kg/day (median of 0.021 mg/kg/day) was slightly higher than that of Lebanese (0.250 mg/g) [56], Kuwaiti (0.280–0.220 mg/g) [65], and Spanish children (0.26 mg/g–before the administration of mouth wash) [66]. In general, fluoride intake depends on the fluoride content of water and water-based beverages [67, 68]. Walia et al. (2017) reported that 68.2% of bottled water is produced locally and contains about 0.07 mgF/l [68], a value 10 times lower than that recommended by US Public Health Service (USPHS), which is 0.7 mg/l [69]. Moreover, in children, tea ingestion could be a good source of fluoride as well as the ingestion of fluoridated toothpaste. Fluoride ingestion was below 0.05 mg/kg body per day and it still remains to be understood whether this low level of fluoride (ingestion or in water) is involved in the development of dental caries, that is highly (~50%) common among UAE children [70, 71]. Dental caries is a common multifactorial oral disease that can affect different age groups [70], and fluoride was found to be effective in reducing the prevalence of this condition [69]. It is reported that 'fluoride in saliva and dental plaque works to prevent dental caries primarily through topical-remineralization of tooth surfaces' [69].

In conclusion, this is a pioneer study that assessed the levels of mineral excretion among the Emirati children. This paper targeted some of the most essential minerals in childhood that can affect growth and development in later stages of life. To the best of our knowledge, no study was done in the UAE at this scale. Several publications address mineral status through questionnaires, which are known to be confounded by several factors including variation in minerals content and their bioavailablity. Hence, there is a gap in the literature about body mineral metabolism and excretion. This research attempts to contribute to the UAE national initiatives to promote preventive measures, enhance general health and reduce the occurrence of certain disorders. The conclusions from this paper set national reference standards for the excretion of Ca, Mg and P among the Emirati youths, and highlighted consumption of iodine and fluoride using biological fluid (urine), elements to be focused on and targeted in future health strategies. We recommend that monitoring studies be carried out to routinely assess the status of these minerals among the UAE population. Collecting additional information about vitamin D, parathyroid hormone, thyroid function, and the level of dental caries can be of an added value to better explain and understand future results.

## Acknowledgments

This study was completed as a result of the invaluable support of the dedicated contributors. We want to thank everyone who facilitated this study namely Zayed university, the ministry of education and the ministry of health and prevention. A special appreciation goes for the help of Ms. Marie-Elizabeth Ragi from the Nutrition Research Laboratory at the AUB for the technical help and support. We also thank the participating students and their families for taking part in the study.

## Author Contributions

**Conceptualization:** Omar Obeid, Dalia Haroun.

**Formal analysis:** Rola Al Ghali, Carla El-Mallah, Linda Smail.

**Funding acquisition:** Dalia Haroun.

**Methodology:** Rola Al Ghali, Ola El-Saleh, Dalia Haroun.

**Project administration:** Omar Obeid.

**Supervision:** Omar Obeid, Dalia Haroun.

**Writing – original draft:** Rola Al Ghali.

**Writing – review & editing:** Rola Al Ghali, Carla El-Mallah, Omar Obeid, Ola El-Saleh, Dalia Haroun.

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
