## [Decision Letter · Decision Letter 0]

10 Jun 2021

PONE-D-21-12286

Urinary minerals excretion among primary schoolchildren in the United Arab Emirates: Alarming iodine levels.

PLOS ONE

Dear Dr. Haroun,

Thank you for submitting your manuscript to PLOS ONE. After careful consideration, we feel that it has merit but does not fully meet PLOS ONE’s publication criteria as it currently stands. Therefore, we invite you to submit a revised version of the manuscript that addresses the points raised during the review process. Specifically, please address the reviewers' comments related to the statistical analysis of data. Further, the study conducted in Dubai should not be generalized to the entire UAE and the title has to be modified accordingly. 

We look forward to receiving your revised manuscript.

Kind regards,

Samson Gebremedhin, PhD

Academic Editor

PLOS ONE

Additional Editor Comments:

Table 1: rather than plain BMI, please compare the BMI for age z-score between boys and girls.

Table 4: for significant ANOVA tests, please also report the results of a post-hoc test.

Journal Requirements:

2. In the ethics statement within the Methods section, please clearly indicate that written informed consent was obtained from the parents/guardian of the minors included in the study.

Furthermore, in your Methods section, please provide a justification for the sample size used in your study, including any relevant power calculations (if applicable).

Reviewers' comments:

Reviewer's Responses to Questions

**Comments to the Author**

1. Is the manuscript technically sound, and do the data support the conclusions?

Reviewer #1: Yes

Reviewer #2: Yes

2. Has the statistical analysis been performed appropriately and rigorously? 

Reviewer #1: Yes

Reviewer #2: No

3. Have the authors made all data underlying the findings in their manuscript fully available?

Reviewer #1: Yes

Reviewer #2: Yes

4. Is the manuscript presented in an intelligible fashion and written in standard English?

Reviewer #1: Yes

Reviewer #2: Yes

5. Review Comments to the Author

Reviewer #1: This is a simple study of the levels of Ca, Mg, I, and Fl in the urine of children in the UAE. The paper is well-organized and well-written.

The methodology is sound, with a large sample size and good precision on measurements.

For the analysis of age effects on mineral excretion, I recommend a correlation analysis instead of dividing into arbitrary age groups, since age is a continuous variable.

The discussion of iodine, including the history of iodine fortification, is very good, and makes important conclusions that iodine fortification may need to be somewhat reduced for optimal health. However, it is important to note that urinary iodine is useful as a good simple measure for populations, not for individuals, due to daily fluctuations.

The discussion of fluoride levels and dental caries is also important.

The study would have been more powerful if thyroid status were evaluated, and if the level of dental caries were assessed, to determine linkages of to iodine and fluoride, respectively.

In summary, this is a simple study that was conducted well, and resulted in important findings on levels of iodine and fluoride in children in UAE.

Reviewer #2: In this study, the urinary concentration of some minerals, including Ca, Mg, P, iodine and fluoride, among children in the United Arab Emirates have been first reported. These results are of great significance for the mineral nutrient assessment in this country. However, there are some issues need to be addressed before it would be published.

1.In 223-231, the iodine status of the Emirati school-aged children was evaluated in this study. However, according to the recommendation of the international organization, median UIC is a population indicator, which only defines the iodine status of a population. It can’t be used to evaluation the individual iodine status.

2.Do the minerals in urine in school-aged children follow the normal distribution? If they did not, the results should be presented as median and interquartile range. As far as I know, at least the UIC does not follow the normal distribution. Besides, the corresponding statistical methods should also be changed.

3.In “Study population and sampling”, the participants were only recruit from Dubai, however, in the title, it showed as “in the United Arab Emirates”. Can children in Dubai represent the whole United Arab Emirates? It is suggested to be corrected as Dubai in the United Arab Emirates.

4.Why is it an “Alarming iodine levels” in the title? In the new recommendation of international organizations, median of UIC in the range of 100~300 are classified as an adequate level. In this study, what is the value of median UIC?

6. PLOS authors have the option to publish the peer review history of their article (what does this mean?). If published, this will include your full peer review and any attached files.

Reviewer #1: No

Reviewer #2: No

---

## [Author Response · Author response to Decision Letter 0]

1 Jul 2021

Additional Editor Comments:

1.Table 1: rather than plain BMI, please compare the BMI for age z-score between boys and girls.

Response: BMI z-score was reported in table 1.

2. Table 4: for significant ANOVA tests, please also report the results of a post-hoc test.

Response: post-hoc test results were added in the results section (line 193-199) and (line 214-218).

3. In the ethics statement within the Methods section, please clearly indicate that written informed consent was obtained from the parents/guardian of the minors included in the study.

Response: This was added in the methodology section (line 98).

4. Furthermore, in your Methods section, please provide a justification for the sample size used in your study, including any relevant power calculations (if applicable).

Response: We targeted 5% of the total population of primary school students (11,147) accounting for approximately 50% response rate and using proportional representation according to school size (lines 86-87).

Reviewer #1: 

1. For the analysis of age effects on mineral excretion, I recommend a correlation analysis instead of dividing into arbitrary age groups, since age is a continuous variable.

Response: In the results section, opening statements were added for tables 4 and 5 that state the results of the correlation between age, as continuous variable, and the different excretions (lines 188-192; lines 212-213). This was followed by the post-hoc results as requested by the editor. The age groups were used to: 1. Compare our findings with other studies who used similar categories; 2. To be able to detect the trends of minerals excretion as children grow and approach pubertal ages. 

2. The discussion of iodine, including the history of iodine fortification, is very good, and makes important conclusions that iodine fortification may need to be somewhat reduced for optimal health. However, it is important to note that urinary iodine is useful as a good simple measure for populations, not for individuals, due to daily fluctuations.

Response: changes in the both the result (line 239-241) and discussion (lines 322-326) sections were done to reflect the population reference rather than the emphasis on the individual classification. 

3. The discussion of fluoride levels and dental caries is also important.

Response: a section about the relationship between fluoride and dental caries was added in the discussion section (lines 364-367).

4. The study would have been more powerful if thyroid status were evaluated, and if the level of dental caries were assessed, to determine linkages of to iodine and fluoride, respectively.

Response: A statement was added as a recommendation in the conclusion section (line 379-383)/

Reviewer #2: 

1.In 223-231, the iodine status of the Emirati school-aged children was evaluated in this study. However, according to the recommendation of the international organization, median UIC is a population indicator, which only defines the iodine status of a population. It can’t be used to evaluation the individual iodine status.

Response: changes in the both the result (line 239-241) and discussion (lines 322-326) sections were done to reflect the population reference rather than the emphasis on the individual classification.

2.Do the minerals in urine in school-aged children follow the normal distribution? If they did not, the results should be presented as median and interquartile range. As far as I know, at least the UIC does not follow the normal distribution. Besides, the corresponding statistical methods should also be changed.

Response: Before checking the normality of all minerals, we were expecting non-normal distribution for Iodine. Surprisingly, this was not the case, Iodine was found to be normally distributed. The skewness and kurtosis were both around 0 (0.6 and 0.2 respectively). In addition, since N is large enough, we assumed normality using the central limit theorem. We included both the median (interquartile range) and mean levels of Iodine in the discussion section to allow for comparisons with other studies (lines 322-323). 

3.In “Study population and sampling”, the participants were only recruit from Dubai, however, in the title, it showed as “in the United Arab Emirates”. Can children in Dubai represent the whole United Arab Emirates? It is suggested to be corrected as Dubai in the United Arab Emirates.

Response: This was changed across the manuscript to Dubai – UAE: in the abstract, methodology and conclusion.

4.Why is it an “Alarming iodine levels” in the title? In the new recommendation of international organizations, median of UIC in the range of 100~300 are classified as an adequate level. In this study, what is the value of median UIC?

Response: The mean value of the population was used to conclude the status of the general study group rather than individual categorization. The Emirati population had a mean of 224 µg/dl and median of 208 µg/dl, classified as at slight risk. Hence, title of the manuscript has been modified and “alarming iodine levels” removed and this was also removed from the conclusion.

---

## [Editor Report · Decision Letter 1]

12 Jul 2021

Urinary minerals excretion among primary schoolchildren in Dubai - United Arab Emirates

PONE-D-21-12286R1

Dear Dr. Haroun,

We’re pleased to inform you that your manuscript has been judged scientifically suitable for publication and will be formally accepted for publication once it meets all outstanding technical requirements.

Kind regards,

Samson Gebremedhin, PhD

Academic Editor

PLOS ONE
---

## [Editor Report · Acceptance letter]

28 Jul 2021

PONE-D-21-12286R1 

Urinary minerals excretion among primary schoolchildren in Dubai - United Arab Emirates 

Dear Dr. Haroun:

I'm pleased to inform you that your manuscript has been deemed suitable for publication in PLOS ONE. Congratulations! Your manuscript is now with our production department. 

Kind regards, 

on behalf of

Dr. Samson Gebremedhin 

Academic Editor

PLOS ONE